# Attenuation of a Field Strain of Infectious Laryngotracheitis Virus in Primary Chicken Culture Cells and Adaptation to Secondary Chicken Embryo Fibroblasts

**Victor A. Palomino-Tapia** *,†,‡ 🅸🄳, **Guillermo Zavala** § 🅸🄳, **Sunny Cheng and Maricarmen Garcia** 🅸🄳

Poultry Diagnostic and Research Center (PDRC), Department of Population Health, College of Veterinary Medicine, University of Georgia (UGA), Athens, GA 30602, USA; avianhealth@gmail.com (G.Z.); scheng@uga.edu (S.C.); mcgarcia@uga.edu (M.G.)
* Correspondence: victor.palominotapia@ucalgary.ca
† This article is part of the Master's thesis from the first author, Victor A. Palomino-Tapia.
‡ Current address: Maple Leaf Foods, 70 Heritage Drive, New Hamburg, ON N3A 2J4, Canada.
§ Current address: Avian Health International, LLC, Flowery Branch, GA 30452, USA.

**Abstract:** The establishment of commercial infectious laryngotracheitis virus (ILTV) live-modified vaccines has relied on serial passaging in chicken embryo (CEO) and tissue culture (TCO) for attenuation. The objective of this study was to attenuate and adapt a virulent CEO-related ILTV field strain (6340) in immortalized cells (LMH), primary chicken embryo kidney cells (CEK), chicken embryo liver cells (CEL), and chicken embryo fibroblasts (CEF). CEFs were refractory to parent ILTV, LMH cells produced low virus yields (~2.5 $\log_{10}$ $TCID_{50}$ per mL), while CEK and CEL cells produced higher viral titers ($\geq \log_{10}$ 6.0 $TCID_{50}$ per mL). After 52 passages in CELs, the cytopathic effect (CPE) was observed not only in hepatocytes but also in CEL fibroblasts. Once CPE was evident in CEL fibroblasts, 20 further passages in CEFs with viral titers reaching yields of ~4.4–5.5 $\log_{10}$ $TCID_{50}$ per mL were performed. The attenuation of CEF-adapted viruses was evaluated after intra-tracheal and conjunctival inoculation in 28-day-old broilers by assessing clinical signs at five days post-inoculation (DPI). Virus CEL cells passages 80, 90, and 100, and CEF passages 10 and 20 were significantly attenuated compared to the parental strain. This is the first report of the attenuation of a virulent field CEO-related ILTV strain (RFLP Group V) in CEF cells—a cell type from a different embryonic germ layer (mesoderm) than ILTV target cells—the respiratory epithelium (endoderm). This finding underscores the potential use of CEF adaptation for the development of a live-attenuated ILTV vaccine.

**Keywords:** infectious laryngotracheitis; TCO; CEO; attenuation; cell culture; serial passage; chicken embryo fibroblasts; CEF

## 1. Introduction

Infectious laryngotracheitis (ILT) is an important respiratory disease of chickens caused by an alpha herpesvirus, genus Iltovirus [1]. ILT outbreaks can cause substantial economic losses due to drops in egg production, weight loss, and mortality [2]. The disease has been historically controlled by biosecurity and vaccination [1].

To present, the following two main types of ILTV vaccines are commercially available: live-modified and recombinant vaccines. Live-modified vaccines have been attenuated by sequential passaging in either embryonated eggs (known as chicken embryo origin or CEO vaccines) [3] or in tissue culture (known as tissue culture origin or TCO) [4], as well as recombinant vaccines with inserts for one or two disease agents, using a backbone of the herpesvirus of turkeys (HVT-LT) or fowl pox virus (FPV-LT) [5]. Currently, a new generation of gene-deleted live ILTV vaccines is being developed [6]. The CEO vaccine is less attenuated than TCO; however, it provides stronger immunity in layers [7]

and broilers [8] and can be administered by mass application via the drinking water or coarse spray routes in addition to eye-drop route, while the TCO vaccine is limited to the individual eye-drop application, and recombinants to *in ovo* or subcutaneous route [9].

Live-modified vaccines (i.e., CEO and TCO) have important drawbacks, including the potential to spread horizontally, revert to virulence, recombine with other ILTVs circulating in the field, and establish latency in the trigeminal ganglia, reactivating later in life and causing disease outbreaks once the immunity wanes [10]. These drawbacks increase the likelihood of the emergence of novel virulent ILTV strains, which later might become the predominant challenge virus in the field [1]. Thus, recombinant vaccines that can be mass-delivered (i.e., *in ovo* day-old vaccination), do not produce ILTV latency, lack vaccine reactivity, and do not revert to virulence were developed. These vaccines protect against mortality and clinical signs; however, they also allow the challenge virus to be shed at similar levels as if vaccinated birds were naïve birds [7,8]. Ranked by protection, ILTV vaccines can be listed as follows: (1) CEO vaccination provides the longest and strongest immunity; (2) TCO vaccine; (3) HVT-LT subcutaneously at day of age; (4) HVT-LT *in ovo*; (5) FPV-LT subcutaneously at day of age; and (6) FPV-Lt *in ovo* [2,7,11].

Recently, some CEO-revertants seem to be causing severe and moderate outbreaks despite vaccination with recombinant vaccines (i.e., HVT-LT) in ILT-endemic areas where CEO vaccines are no longer permitted (e.g., British Columbia [BC]—Canada) [11]. Control of the disease in these areas has proved difficult, as, due to the ban on CEO vaccines, the inadequate application of the TCO vaccine is common (i.e., a double dose via drinking water vaccination). This out-of-label usage of the TCO vaccine may have facilitated the emergence of TCO-CEO recombinants isolated from clinical cases of unknown virulence in BC, Canada [11]. Thus, the search for safer vaccines against ILTV should include the following: the ability to be mass-applied (e.g., *in ovo* or at day of age), lack of virulence reversion, low viral shedding upon field challenge, and the ability to control high virulent field ILTV outbreaks such as the ones occurring in Canada [12]. Furthermore, attenuation on a cell type (i.e., CEF) from a different embryonic germ layer origin than the target cells (i.e., tracheal epithelium) might provide additional safety measures in vaccine development. The differential host-cell signaling [13,14], tissue-specific factors-receptors [14–16], and altered immune responses [15,17] resulting from attenuation in CEF might collectively contribute to attenuating the parental virus (i.e., allowing for *in ovo* application), while preserving immunogenicity and reducing unwanted events, such as strong vaccine reactions, recombination, and reversion to virulence.

The objective of this study is to attenuate a field strain of ILTV in primary cell cultures (i.e., CEL and CEK) as previously described [4,18], and its further adaptation to chicken embryo fibroblasts (CEF) to increase its attenuation for the development of a safe, non-revertant vaccine that can be mass-applied under commercial conditions (e.g., *in ovo*).

## 2. Materials and Methods

### 2.1. Virus Strain

The 63140 ILTV field strain used in the present experiment was isolated in chicken kidney (CK) cells from an outbreak in 45-day old broilers in Georgia, USA [7]. Later, 63140 was typed as a virulent CEO-related genotype group V strain of ILTV, responsible for severe clinical signs and mortality in naive broiler chickens [2]. The passage history of the for the viruses that were evaluated for attenuation are shown in Table 1. An eighth passage (P8) of 63140 in CK cells was used as a parental virus for the serial passages for CEK and CEL serial passages, and as a challenge strain in subsequent challenge experiments.

**Table 1.** Stock titrations expressed in TCID$_{50}$ per mL by Reed–Muench with 95%CI (modified Kärber). Different letters indicate statistically significant differences ($p \leq 0.05$) within each type of tissue culture (i.e., CEK, CEL, and CEF).

| PASSAGE | CEK | CEL | CEF[y] (Started from CEL-P52) |
|---|---|---|---|
| 06 | ND [b] | ND [b] | log$_{10}$ 4.38 (log$_{10}$3.95–4.65) [a] |
| 10 | log$_{10}$ 6.50 (log$_{10}$6.50–6.50) [d] | log$_{10}$ 6.38 (log$_{10}$5.95–6.65) [c] | log$_{10}$ 5.50 (log$_{10}$5.50–5.50) [b] |
| 20 | log$_{10}$ 4.83 (log$_{10}$4.47–5.32) [b] | log$_{10}$ 6.63 (log$_{10}$6.35–7.05) [c] | log$_{10}$ 4.50 (log$_{10}$4.50–4.50) [a] |
| 30 | log$_{10}$ 3.17 (log$_{10}$2.67–3.52) [a] | log$_{10}$ 5.83 (log$_{10}$5.47–6.33) [bc] | ND [z] |
| 40 | log$_{10}$ 6.63 (log$_{10}$6.34–7.05) [d] | log$_{10}$ 6.50 (log$_{10}$6.50–6.50) [c] | ND [z] |
| 50 | log$_{10}$ 5.83 (log$_{10}$5.47–6.33) [cd] | log$_{10}$ 5.63 (log$_{10}$5.35–6.05) [b] | ND [z] |
| 60 | log$_{10}$ 5.17 (log$_{10}$4.67–5.53) [bc] | log$_{10}$ 5.68 (log$_{10}$5.15–6.26) [b] | ND [z] |
| 70 | ND [z] | log$_{10}$ 5.38 (log$_{10}$4.95–5.65) [ab] | ND [z] |
| 80 | ND [z] | log$_{10}$ 5.63 (log$_{10}$5.35–6.05) [bc] | ND [z] |
| 90 | ND [z] | log$_{10}$ 6.63 (log$_{10}$6.35–7.05) [c] | ND [z] |
| 100 | ND [z] | log$_{10}$ 6.50 (log$_{10}$6.50–6.50) [c] | ND [z] |
| x̄ 95% CI | log$_{10}$ 5.36 (log$_{10}$ 4.01–6.70) | log$_{10}$ 6.08 (log$_{10}$ 5.73–6.43) | log$_{10}$ 4.79 (log$_{10}$ 3.27–6.32) |
| Coefficient of Variation | 8.07% | 23.98% | 12.83% |

ND = Not done.

### 2.2. Primary Chicken Cells and Cell Line

Continuous cell lines derived from chicken hepatocellular carcinoma (LMH) [19], and chicken primary cells, such as chicken embryonic liver (CEL), chicken embryonic kidney (CEK) cells, and secondary chicken embryonic fibroblast (CEF), were used to attempt attenuation of the 63140 ILT strain.

Primary CEL, CEK, and secondary CEF cells were prepared from 10–15, 17–20, and 7–10-day-old SPF embryos, respectively. Growth media (GM) and maintenance media (MM) were used to perform the cell cultures. GM consisted of DMEM (Mediatech Inc., Manassas, VA, USA) with 10% fetal bovine serum (FBS) and 2% antibiotic-antimycotic (Ab) solution 100×, while MM consisted of DMEM with 1% calf serum (CS) + 2% Ab. Briefly, in the case of CEL and CEK, organs were harvested, minced, and washed three times in 1× PBS to eliminate red blood cells as much as possible. Cells were then placed in a 50 mL trypsinizing flask with pre-warmed trypsin with EDTA (Mediatech Inc., Manassas, VA, USA) for a 10 min trypsinization step. Thereafter, cold GM was added to stop trypsinization, and cells were filtered through sterile cheesecloth before centrifugation at 410× *g* for 10 min. The cell pellet was then resuspended in pre-warmed GM, at a cell density of $5 \times 10^5$ cells per mL and plated in T25 tissue culture flasks. Flasks were incubated at 37 °C in an atmosphere enriched with 5% CO$_2$ until reaching 85–95% confluence. The same procedure was used for CEF, except that the cells were cultured in a T75 culture flask and trypsinized when the monolayer reached 100% confluency prior to sub-culturing in T25 flasks to obtain secondary CEF.

The immortalized cell line LMH was kindly provided by Dr. Garcia's laboratory at the University of Georgia and consisted of a subset of a culture originally acquired from American Type Culture Collection (ATCC) and adapted to multiply at 39 °C. Subcultures were performed according to ATCC recommendations. Thus, in case of LMH cells, the HyClone® DME/F-12 media (HyClone Laboratories Inc., South Logan, UT, USA) was used instead of DMEM for GM and MM.

### 2.3. Serial Passages

Two T25 flasks with 85–95% confluent monolayers were used to perform each passage. One flask was mock-inoculated with sterile MM, while the second flask was inoculated with the supernatant containing the viral strain obtained from the previous passage. After inoculation, monolayers of CEF, CEL, and CEK were absorbed from 1 to 2 h at 37 °C, while LMH cell monolayers were absorbed from 1 to 6 h at 37–39 °C. After adsorption,

5 mL of pre-warmed MM was placed into each flask and CEL, CEK, CEF monolayers were incubated at 37 °C with 5% $CO_2$ for five days, while LMH inoculated cells, were incubated at 39 °C with 5% $CO_2$ for five days. At passage levels CEL-P60 and CEF P12, the incubation period was reduced from 5 to 2 days to increase the number of passages, and thus accelerate the attenuation process.

After incubation, both flasks were frozen and thawed three times. Cultures were then centrifuged and a 200 μL aliquot from each culture supernatant was inoculated into a fresh cell culture 85–90% confluent monolayer with for a subsequent passage.

At every tenth passage in CEL, CEK, CEF, and at the eight LMH passage, supernatants were titrated in chicken kidney (CK) cells from 3- to 4-week-old SPF chickens in 96-well plates as previously described [8]. In short, a typical assay was a setup of 5 replicates at each concentration of the supernatant containing the ILTV passage. The inoculum used in the first row was a tenth of the concentration of the stock, and each following row of replicates was diluted 10-fold. Virus titers were determined by the Reed and Muench method [20], and statistical ranges of 95% confidence intervals were determined using modified Kärber method [21].

### 2.4. Conventional PCR, Real-Time PCR, and Reverse Transcriptase (RT)-PCR

To exclude potential contamination of the continuously passaged cultures RT-PCR and PCR assays for detection of Avian Reovirus, Fowl Adenovirus, and Mycoplasmas were performed after every 20 passages by RT-PCR or PCR. Total DNA and RNA were extracted from infected cells using a commercial extraction kit for DNA (High Pure Template DNA Purification Kit, Roche Diagnostics, Indianapolis, IN, USA) and RNA (High Pure Template RNA Purification Kit, Roche Diagnostics, Indianapolis, IN, USA) following the manufacturer's instructions. The conserved region of the Sigma C gene was amplified for avian reovirus [22], the L1 Loop of the hexon protein gene was amplified for fowl adenovirus [23], and the 16S and 23S rRNA genes of *Mycoplasma* spp. were amplified [24]. RT-PCR and PCR conditions were performed as previously described. In the case of ILTV, the glycoprotein B (gB) gene of approximately 2.7 kb was amplified. The gB PCR was performed in a final reaction of 20 μL that consisted in 18 μL of Platinum PCR Super Mix High Fidelity (Invitrogen, Carlsbad, CA, USA), primers combined to a final concentration of 1 μM and 1 μL of DNA template. The thermal profiling used was as follows: 94 °C for 2 min, 35 cycles of 94 °C for 30 sec, 55 °C for 45 sec, and 68 °C for 3 min, and a final cycle of 68 °C for 10 min.

ILTV genome load in tracheal swabs was quantified by real-time PCR (qPCR) as previously described [8]. Tracheal viral loads were assessed by qPCR on tracheal swabs at 5 days post-inoculation (DPI) in the CEL-P100, CEF-P20, 63140 Parental, and Negative Control groups. Tracheal viral loads were expressed as $Log_{10}$ ($2^{-\Delta\Delta Ct}$) genome copy numbers (GNC). DNA extraction from tracheal swab samples was performed using the MagaZorb® DNA mini-prep 96-well kit (Promega, Madison, WI, USA) following manufacturer's recommendations with modifications as previously published [7]. A summary of the primers utilized to detect ILTV, Avian Reovirus, Fowl Adenovirus, and *Mycoplasma* spp. are shown in Supplement Table S1.

### 2.5. Bacteria Sterility of Tissue Culture Passages

Every 10 passages, all tissue cultures were examined for bacteria contamination by culture on trypticase-soy agar (TSA) (Oxoid USA, Inc., Columbia, MD, USA) at 37 °C for up to 48 h under aerobic conditions [25]. Mycoplasma culture was performed only upon unexpected changes in infected cell monolayers (strange CPE in fibroblasts of infected CEL cells at P52). Mycoplasma culture was performed at 37 °C for up to 5 days using in-house Frey's medium as previously published [25].

## 2.6. Preparation of Hyperimmune Sera and Indirect Immunofluorescent Antibody Test

Polyclonal hyperimmune serum against 63140 ILT was produced in four-week-old specific pathogen-free (SPF) chickens. Briefly, SPF chickens were inoculated three times at two-week intervals with the 10th 63140 CEL passage virus. The first inoculation at 4 weeks of age was through the conjunctiva (50 μL in each eye), the intranasal route (50 μL in each nostril), and intramuscular route (400 μL for each breast). The second and third inoculations were only given by intramuscular route in the breast (0.5 mL for each breast). Fifteen days after the final inoculation, the chickens were bled, and the serum was harvested and frozen at −20 °C until further use.

Indirect immunofluorescence was performed in ILTV-infected monolayers seeded in 96-well plates. Briefly, media was removed, and cells were washed with warm 1× PBS, cells then were fixed with for 20 min with cold 100% cold ethanol. Cells were dried at room temperature (RT), 5% skim milk in 1× PBS blocking solution was added per well, and plates were incubated at 37 °C for one hour. Plates were washed three times at RT (1× PBS), incubated for hour with ILTV chicken antiserum (1:10 dilution) at 37 °C, and followed by three additional washes with 1× PBS. A second 1 h incubation at 37 °C was performed with a 1:200 dilution of FITC-labeled secondary mouse anti-chicken IgG antibody (Sigma, Saint Louis, MS, USA), followed by three washes as described above. Finally, 100 μL of DABCO-glycerol as mounting solution (2.5 mg/mL DABCO (1.4 diazobicyclo-[2.2.2.]-octane, Sigma-Aldrich, St Louis, MO, USA) in 90% [*v/v*] glycerol (Merck, Kenilworth, NJ, USA) in 1× PBS) was further diluted 1:1 in 1× PBS and added to each well to prevent photobleaching. Finally, stain cells were examined under the microscope (Olympus IX81, Olympus corporation, Center Valley, PA, USA).

## 2.7. Electron Microscopy-Negative Stain Technique

A formvar, carbon-coated 400-mesh copper grid was floated on a 40 μL drop of the tissue culture supernatant submitted to the Electronic Microscopy facility at the College of Veterinary Medicine at the University of Georgia. The grid was removed from the drop and the excess was wicked off with the edge of clean filter paper. The grid was floated on a drop of 3% aqueous phosphotungstic acid, pH 6.8–7.0, for 30 s. After wicking off the excess stain from the grid with the edge of clean filter paper, the grid was allowed to dry on filter paper before viewing with the transmission electron microscope (TEM). The TEM used was a JEOL JEM-1210 and was used at an accelerated voltage of 120 kv [26].

## 2.8. Experimental Design to Evaluate Attenuation of CEL and CEF 63140 Passages

A total of 150 unvaccinated broiler chickens were acquired from a commercial source at hatch, and reared in filtered-air isolation units at the Poultry Diagnostic and Research Center (PDRC, University of Georgia, Athens, GA, USA) with feed and water ad libitum. At 28 days of age, 14 groups of chickens (*n* = 5) with one replicate, were inoculated with 63140 CEL-P10, 20, 30, 40, 50, 60, 70, 80, 90, 100, CEF-P6, P10, P20, and parental virulent 63140 virus at a dose of $\log_{10}$ 3.5 $TCID_{50}$ administered in a volume of 200 μL per chicken (100 μL intratracheal and 50 μL in each eye). One group of chickens, with one replicate, was mock inoculated with MM in a similar fashion as described above and was considered as the negative control.

At five days post-inoculation (DPI), clinical sign categories, such as conjunctivitis, dyspnea, and lethargy, were scored as previously described [8]. Individual categories of clinical signs were scored from 0 to 3, with 0 being normal, 1 = mild, 2 = moderate, and 3 = severe for each of the categories. In detail, conjunctivitis was scored as 0 = normal, 1 = mild (minimal swelling, minimal closure of eyes), 2 = moderate (moderate swelling, partial closure of eyes), 3 = severe (severe swelling, complete closure of eyes). Dyspnea was scored as 0 = normal; 1 = mild (minimal open-mouth breathing with little or no presence of mucus in the larynx); 2 = moderate (moderate open mouth breathing/gasping with moderate presence of clear or bloody mucus in the larynx); 3 = severe (severe gasping with extended neck with severe presence of mucus in the larynx). Lethargy was scored

as 0 = normal; 1 = mild (ruffled feathers, head and neck sunk on the chest when undisturbed, but immediately alert when approached); 2 = moderate (ruffled feathers, head and neck sunk on the chest when undisturbed, not immediately alert when approached); and 3 = Severe (ruffled feathers, head and neck sunk on the chest when undisturbed, no change or minimal change in conduct when approached). The total clinical signs score per chicken was estimated based on the sum of scores for the three categories. Any mortality received a total score of nine. Individual chickens and median total clinical sign score per group were calculated. Tracheal swab samples were collected at 5 DPC from 63140 Parental, CEL-P100, CEF-P20, and Neg Control. Swabs were collected into a 1.8 mL microcentrifuge tube containing 1 mL sterile PBS with 2% antibiotic-antimycotic 100× solution (Gibco, Grand Island, NY, USA) and 2% calf serum re-suspended for 20 s and stored at −80 °C until processing.

*2.9. Statistical Analysis*

The GraphPad Prism version 10.1.0 statistical package (GraphPad Software, La Jolla, CA, USA) was used to analyze clinical sign score data obtained from challenge studies. Virus titers were expressed in $TCID_{50}/mL$ determined using Reed and Muench formula [20]. Standard errors, and 95% coefficient intervals were obtained using the modified Kärber formula [21]. All qPCR and $TCID_{50}/mL$ data were analyzed using one-way ANOVA with Bonferroni's method for multiple pair-wise comparison, while Kruskal–Wallis test was independently used to compare median clinical sign scores for each group against the group inoculated with the parental 63140 strain (positive control), followed by multiple pair-wise comparisons to search for statistical differences as previously published [7].

**3. Results**

*3.1. Serial Passaging in Primary Chicken Cells*

Primary CEK and CEL cells were permissive to the parental ILTV 63140 strain, with CEL tolerating ILTV infection from 5 to 7 days post-inoculation (pi), while CEK cells only as early as from 2 to 4 days pi. Incubation in CEL and CEK monolayers was limited by total detachment of the monolayer from the flask (usually between 0 and 4 DPI) or by infection and cell death of all susceptible cells in the culture and formation of a monolayer formed by ILTV-refractive cells (i.e., CEFs from liver and kidney—usually between 5 and 7 DPI). CPE in CEL and CEK cell cultures was characterized by the formation of multinucleated giant cells, cell degeneration, and necrosis, as shown in Figure 1a for CELs and 1c for CEK cells. Virus yields in CEL, CEK, and CEF cells ranged from $\log_{10}$ 5.38 to 6.63, from 4.83 to 6.63 $\log_{10}$, and from 4.38 to 5.50 $TCID_{50}$ per mL, respectively (Table 1). $TCID_{50}/mL$ Average titers for CEK, CEL, and CEF were $\log_{10}$ 5.36, $\log_{10}$ 6.08, and $\log_{10}$ 4.79. Although numerically different, CEL produced numerically more titers than CEK, and CEK more than CEF, but there was no significant statistical difference between these cell culture systems (Table 1). On the other hand, after inoculation of the 63140 parental strain on LMH cells, CPE was observed for 8 passages, suggesting replication, although titer yield in this cell line was much lower (~$\log_{10}$ 2.5 $TCID_{50}$ per mL), and thus virus passages on LMH cells were not continued. When the parental 63140 strain was first inoculated in CEF, no cytopathic effect was detected. However, from CEL P48 onwards, it was noticed that the derived liver fibroblasts (Figure 1b), hitherto refractive to ILTV infection, were also showing signs of CPE, suggesting that the CEL P48 virus was capable of infecting both hepatocytes and fibroblasts in the CEL culture. To determine if the CEL passage virus was capable of productively infecting fibroblasts, the CEL-P52 virus was inoculated into a secondary CEF monolayer, and CPE was observed as early as 48 h post-inoculation. The CPE consisted mainly of cell rounding and the appearance of vacuoles in the cytoplasm of rounded fibroblasts that formed multicellular aggregates that remained attached to the monolayer (Figure 2a). PCR assays to rule out possible infection with other avian viruses that may cause a similar CPE in fibroblasts, such as Fowl Adenovirus and Avian Reovirus, were performed, and all resulted in negative detection (results not shown). Productive

infection of ILTV in CEF was confirmed by PCR (Figure 3), IFA (Figure 4), and electron microscopy (EM) (Figure 5). CEF-P1 and CEF-P6 were tested for *Mycoplasma* spp. PCR, for Mycoplasma spp. isolation, and for bacteria sterility. *Mycoplasma* spp. DNA was not detected or isolated. Tissue culture supernatants obtained every 10th passage were free from bacteria. At passage levels CEL-P60 and CEF P12, the incubation period was reduced from 5 to 2 days to accelerate the attenuation process by increasing the number of passages.

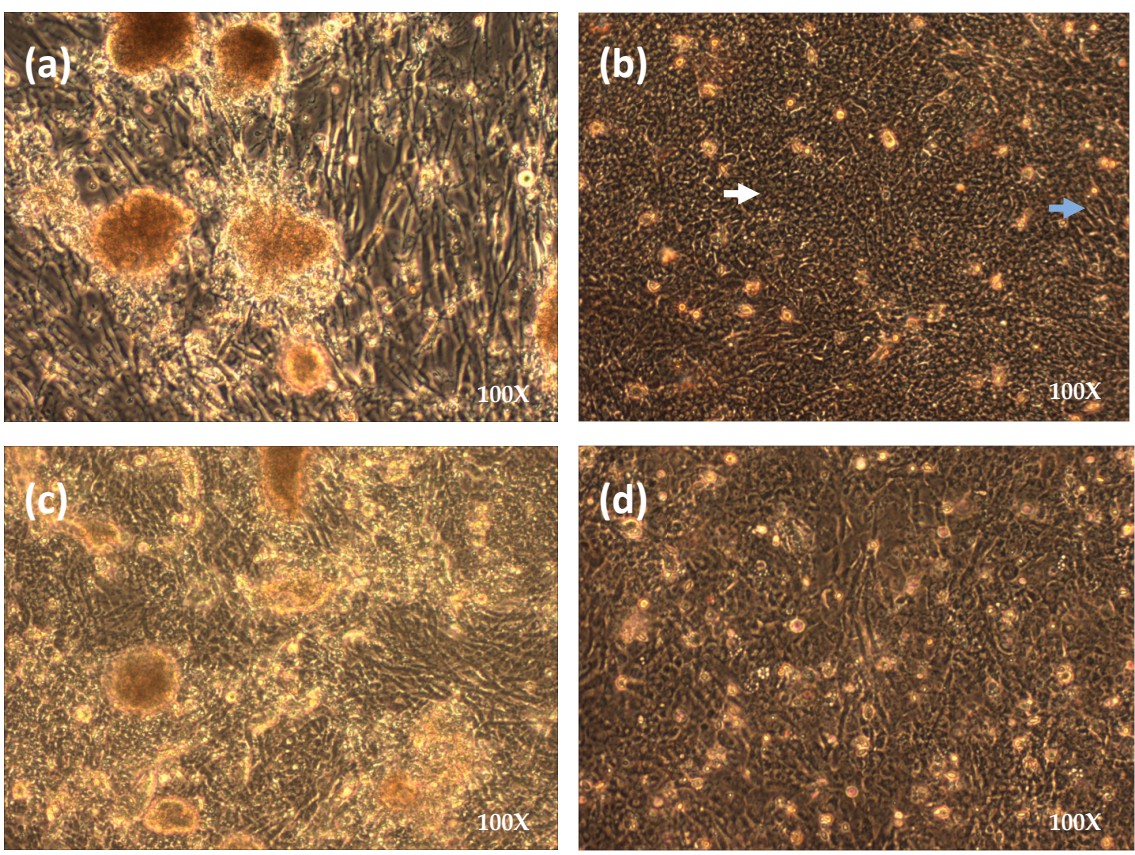

**Figure 1.** CPE produced by 63140 inoculations in CEL and CEK cell cultures observed at 100×. (**a**) CEL cells at 5 days post infection (DPI) with CEL-P20 inoculum. (**b**) CEL control cells—hepatocytes (white arrow), fibroblasts from liver (light blue arrow). (**c**) CEK cells at 2 DPI with 63140 CEK-P22 inoculums. (**d**) CEK control cells.

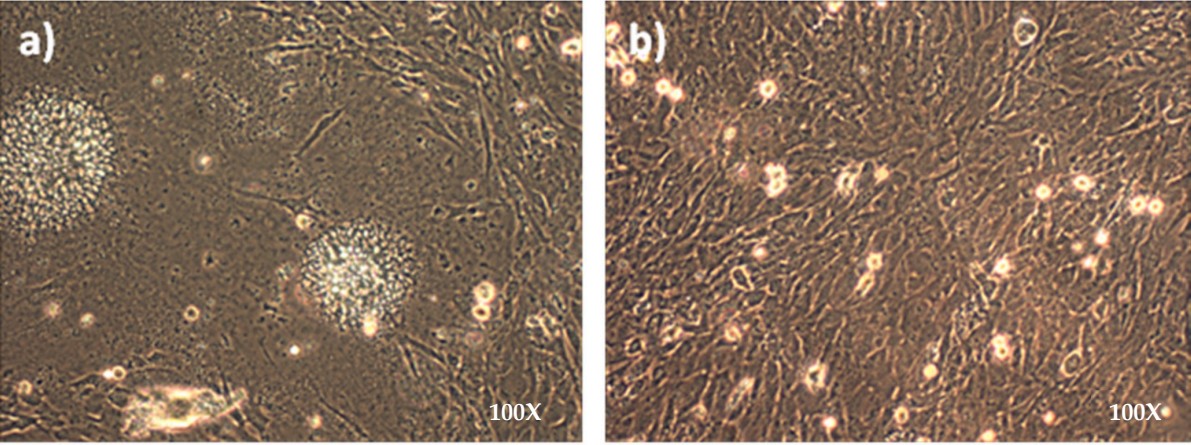

**Figure 2.** CPE produced by 63140 in CEF cell cultures observed at 100×. (**a**) CEF infected with 63140 CEF-P4 at 1.5 DPI. CPE is observed by fusion of fibroblasts forming a multicellular plain aggregate. (**b**) CEF control cells.

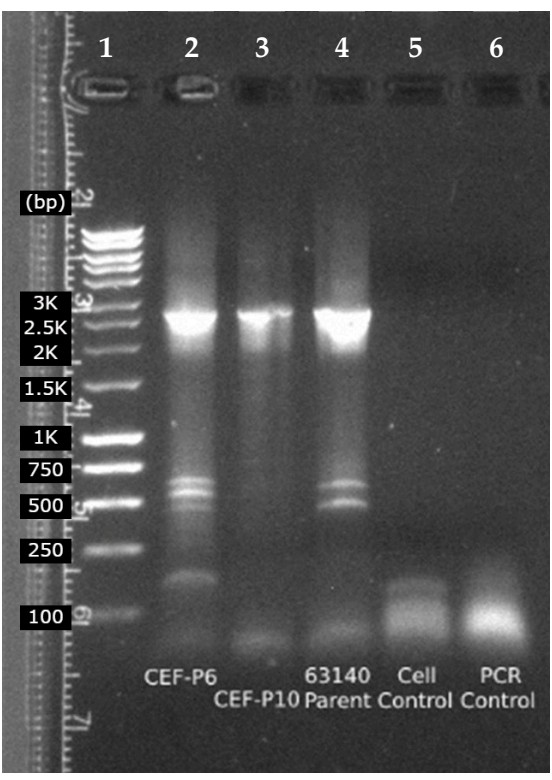

**Figure 3.** PCR analysis of CEF adapted 63140 ILTV strain. A PCR product of 2696 bp is consistent with ILTV glycoprotein B open reading frame amplicon generated by gB-BNCU158 and gB-BNCL2854 (Supplement Table S1). Lane 1 molecular weight marker with relevant band sizes in kilo base pairs (Kbp). Lane 2 to 4 total DNA extracted from CEFs infected with CEF-P6 (2), CEF-P10 (3), 63140 parent (4), and non-infected CEFs cells (5). Lane 6 PCR control, respectively.

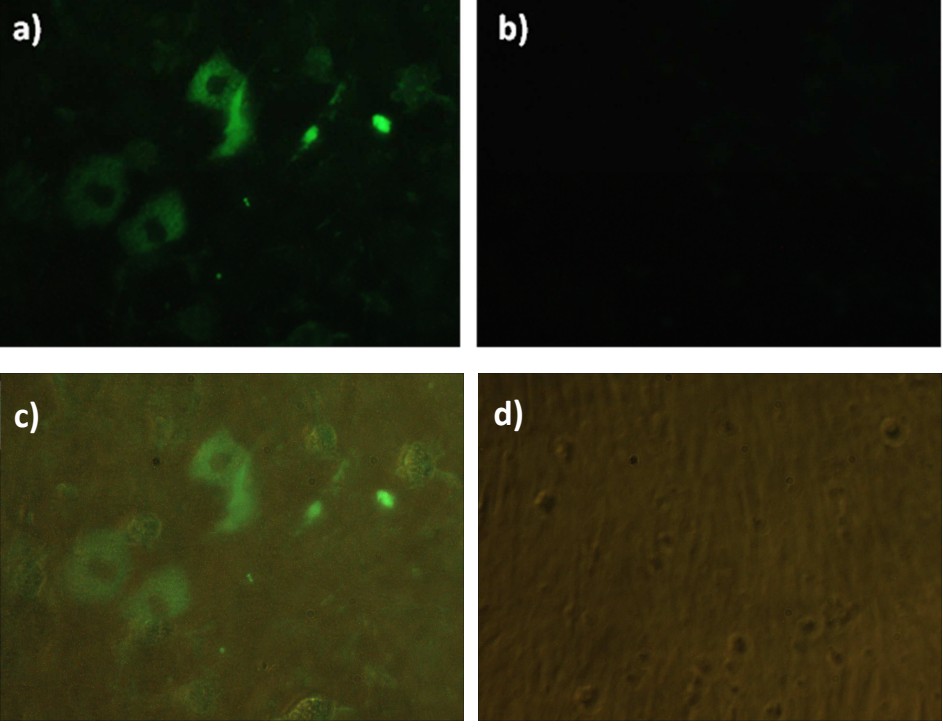

**Figure 4.** Indirect Immunofluorescence assay (IFA) observed at 400×. (**a**) 63140 CEF-P5 at 5 DPI; (**b**) non-infected negative control CEFs at 5 DPI. Bright field images observed at 400× (**c**) 63140 CEF-P5 at 5 DPI; (**d**) non-infected negative control CEFs at 5 DPI.

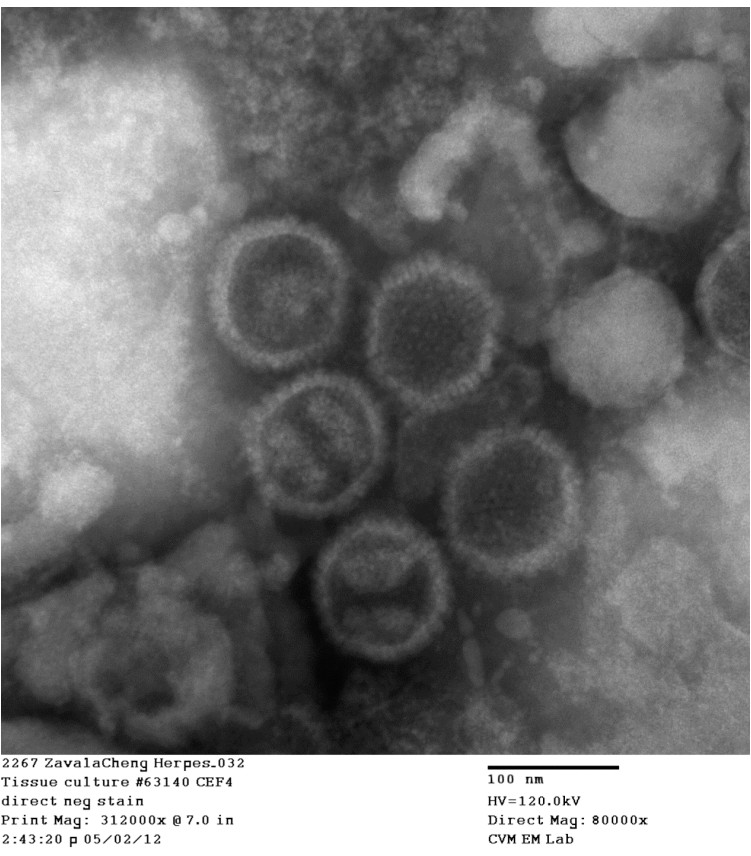

**Figure 5.** Transmission electron microscopy of negatively stained CEF cells infected with the P3 CEF 63140 (amplification = 80,000×). Five viral capsids are observed. Reference bar = 100 nm.

*3.2. Virus Titration*

A summary of stock titrations expressed in $TCID_{50}$ per mL for CEK and CEL is shown in Table 1. Only CEL passages were continued beyond the 60th passage due to the CEL cultures capability to tolerate ILTV infection longer than CEK (5–7 days vs. 2–4 days), and because it generated higher virus yields. Compared to CEL, the titers for the CEF passages 6, 10, and 20 were lower, ranging from 4.38 to 5.5 $\log_{10} TCID_{50}/_{mL}$.

*3.3. Attenuation 63140 Passages in CEL and CEF*

The group of chickens inoculated with the 63140-parent virus (positive control) had the highest median clinical sign score amongst all CEL and CEF treatment groups, whereas the negative control group had the lowest median clinical sign scores amongst all CEL and CEF treatment groups (Figures 6 and 7). The predominant clinical signs observed for the 63140-parent group of chickens were lethargy and respiratory signs such as snicking and gasping due to the presence of obstructions in the trachea (bloody/caseous exudate). Conjunctivitis was mainly observed in the 63140-parent group and was rarely observed in the groups of chickens inoculated with CEL and CEF viral passages as early as CEL-P10. Lethargy and respiratory sign scores for the CEL-P10, CEL-P20, and CEL-P30 groups of chickens were comparable to the 63140-parent group but progressively decreased in later passages (CEL-P40 to CEL80 and CEF-P6 to CEF-10) with a slight numerical increase on CEL-P90, CEL-P100, and CEF-P20. Median clinical sign scores from passages CEL-P80, CEL-P90, and CEL-P100 were statistically different from the 63140 parental-inoculated group (positive control), as shown in Figure 6. While median clinical sign scores from passages CEF-P10 and CEF-P20 were statistically different from the median clinical signs induced by the parental 63140 parental virus, as shown in Figure 7. When analyzing all experimental groups together, attenuation can be observed in groups CEL-P80, CEL-P90, CEL-P100, CEF-P6, CEF-P10, and CEF-P20 (Figure 8).

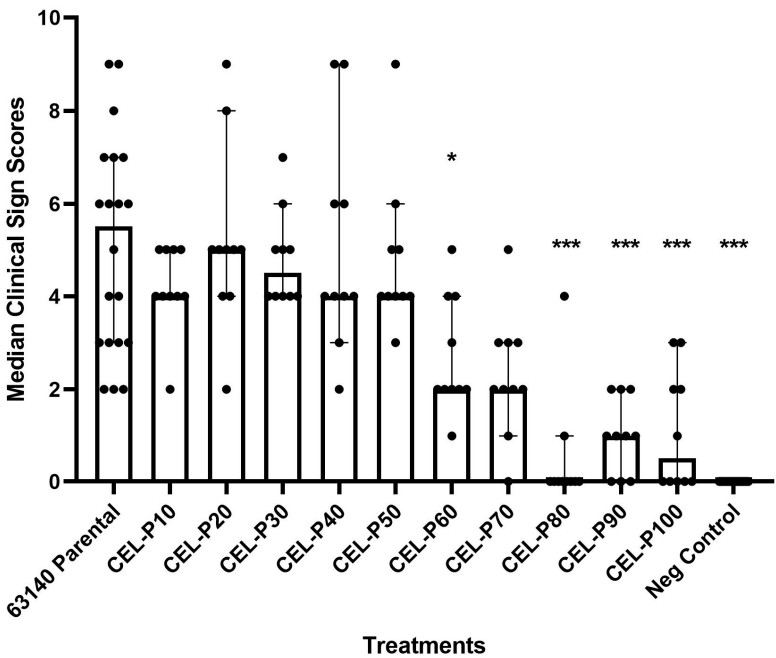

**Figure 6.** Clinical sign scores at 5 days post-inoculation of Safety Trial #1 and 2 on CEL passages. Black dots represent individual values. Asterisks indicate a statistically significant difference relative to the positive control (Ch). Data are presented as median ± 95% CI (*** $p < 0.001$, * $p < 0.05$).

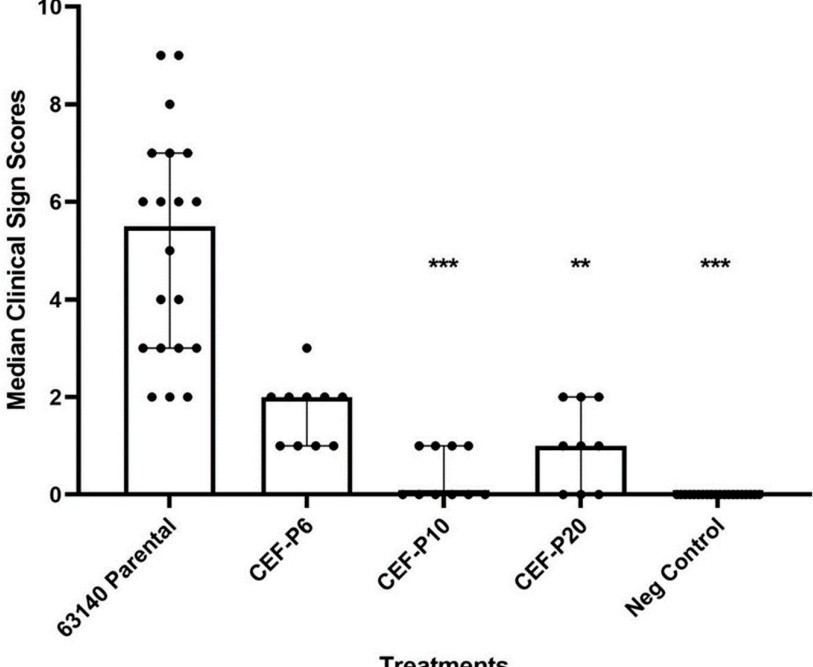

**Figure 7.** Clinical sign scores at 5 days post-inoculation of Safety Trial #1 and 2 on CEF passages. Black dots represent individual values. Asterisks indicate a statistically significant difference relative to the positive control (Ch). Data are presented as median ± 95% CI (*** $p < 0.001$, ** $p < 0.01$).

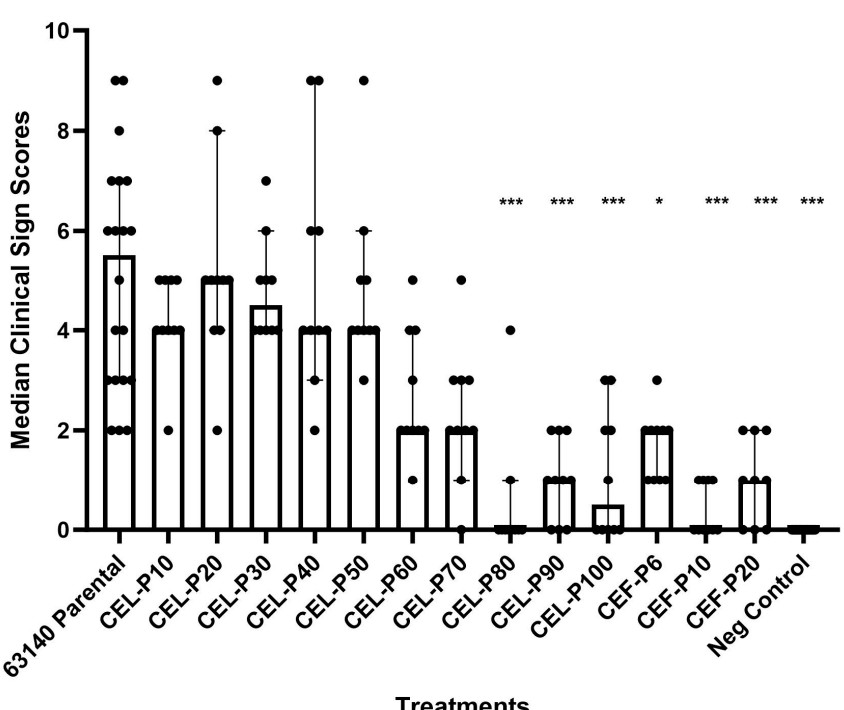

**Figure 8.** Clinical sign scores at 5 days post-inoculation of Safety Trial #1 and 2 on CEL, CEK, and CEF passages. Black dots represent individual values. Asterisks indicate a statistically significant difference relative to the positive control (Ch). Data are presented as median ± 95% CI (*** $p < 0.001$, * $p < 0.05$).

## 4. Discussion

Propagation of ILTV in cell cultures has been well-documented, e.g., Vero [27,28], LMH [29], QT-35 [30], CEF [18,28], CEL, and CEK [27]. In contrast, reports describing in vitro attenuation of a virulent virus in cell culture are scarce [4,31,32].

The present research shows, for the first time, the attenuation in tissue culture and adaptation to CEF of a virulent 63140 field CEO-related ILTV genotyped as group V by RFLP [33] and full genome sequence [34].

Compared to the parental 63140 virus, two new virus lineages were generated. One virus showed a significant degree of attenuation in SPF chickens after 80 continuous passages in CEL, and a second virus showed a significant degree of attenuation and adaptation to CEF after 52 passages in CEL, followed by 10 passages in CEF. Despite the significant decrease in clinical signs induced by a virus with a higher passage level, mild to moderate clinical signs were still detectable in some chickens inoculated with CEL-P90, CEL-P100, CEF-P10, and CEF-P20. We speculate that this tendency in attenuation, where the overall clinical signs plateaued yet still, a few chickens were showing signs of disease, was a consequence of decreasing the culture passage incubation time from 5 to 2 days. The authors speculate that the shorter incubation time selected for ILTV subpopulations that were better fitted to replicate in tissue culture and in vivo resulted in the inoculation affecting the tracheal mucosa and numerically increasing the clinical signs. The viral genome load in the trachea of CEF-P20, CEL-P100, and parental 63140 strains were not significantly different at 5 days post-challenge (Figure 9). Although not directly compared, it seems that CEL-P60 and CEL-P70 would be not attenuated and more virulent than CEF P6 (CEL-P52 + CEF-P6), suggesting that CEF would be a faster cell type for attenuation (Figure 8). Traditionally, viral attenuation has been achieved by serial passage in a permissible cell type from the host [4] or from a foreign host species [35], allowing the virus to adapt to a cell type from the same embryonic origin as the target organ, or foreign host and become less virulent [36]. Relevant examples are the field ILTV attenuation in embryonic hepatocytes of chicken and turkey [37], which, together with the

pseudostratified ciliated epithelium lining the tracheal lumen (target cells for ILTV), share the same embryonic origin—the endoderm germ layer [38]. The present paper shows the first account of attenuation of an ILTV field virus in a cell type from a different embryonic origin than the target tissue (i.e., cross-cell attenuation) as secondary CEFs are derived from the mesoderm and not from the endoderm germ layer [39]. The authors speculate that secondary CEF passaging reduced clinical signs faster than CEL passages due to the following three main mechanisms: (1) Host cell signaling—by passaging the virus in a different cell type, the viral replication machinery may be altered, leading to changes in the viral gene expression, protein synthesis, and host cell interactions [13,39]; (2) tissue-specific factors—a virus adapted to replicate in a cell type of different embryonic germ layer than the target cell may exploit specific factors or receptors that are not abundantly present in the target cell [14–16]. This can alter the viral tropism, limiting its ability to efficiently infect and replicate in the target organ. As a result, it may exhibit reduced virulence without compromising its immunogenic properties; and (3) host immune responses—the resulting attenuation process might alter the viral antigenicity or immunomodulatory properties [15,17]. Thus, the attenuated virus may induce a more robust immune response in the target organ, promoting the generation of protective immunity against subsequent infections while minimizing the risk of severe disease. Any of these mechanisms, or a combination of them, can result in the observed results of our animal studies—a reduced virulence upon infection of the target organ.

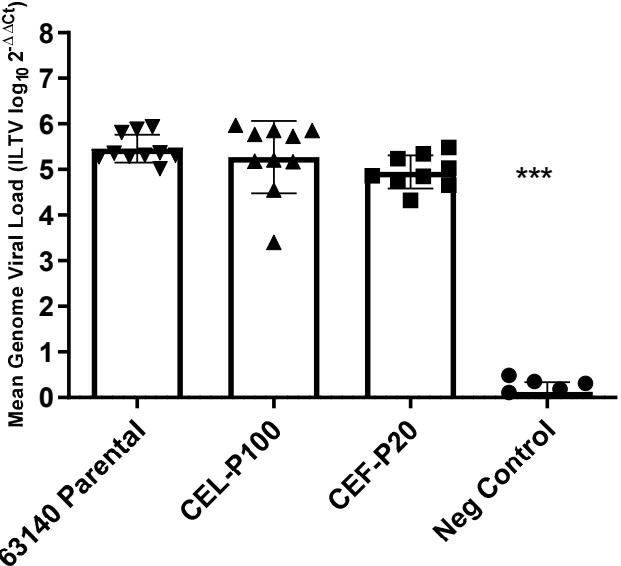

**Figure 9.** Trachea challenge virus load determined by quantitative ILTV-PCR at day 5 post-challenge. The amount of the viral nucleic acid relative to the amount of chicken $\alpha$2-collagen was expressed as $\text{Log}_{10}$ ($2^{-\Delta\Delta Ct}$). At day 5 post challenge, the average $\log_{10} 2^{-\Delta\Delta Ct}$ values were $5 \times 10^{-9}$ (Neg Control); 4.95 (CEF-P20); 5.27 (CEL-P100); and 5.45 (63140 Parental). Inverted black triangles, black triangles, black squares and black circles represent individual values on each treatment group. Asterisks indicate a statistically significant difference relative to the positive control (Ch). Data are presented as Mean $\pm$ SD (*** $p < 0.001$).

This may explain the decrease in clinical sign scores in CEF passages, as observed in Figures 6–8. Further attenuation in CEF may lead to a safe vaccine able to be delivered *in ovo* by reducing the vaccine virus's ability to cause disease in the target organ and adapting the virus to a different cell type (e.g., fibroblasts), which can also be targeted by subcutaneous delivery, rich in fibroblasts susceptible to this CEF-adapted strain. In addition, the use of secondary CEF for the propagation of a tissue culture live-modified ILTV would considerably reduce vaccine production costs as CEL primary cells require the use of older embryos, more labor to collect livers, and low cell yield on monolayers,

whereas secondary CEFs are routinely used for vaccine production of many other viruses, such as Marek Disease virus (MDV) serotype 1 (e.g., Rispens); MDV serotype 2 (i.e., SB-1); MDV serotype 3; or HVT (e.g., FC 126), Infectious Bursal Disease, Avian Reovirus, and Fowl Pox virus. In addition, CEF production requires a younger embryo (which requires less welfare regulation) and produces high cell yields in monolayers, therefore improving and decreasing the costs of vaccine manufacturing [40].

These results demonstrate that attenuation of the ILTV 63140 strain in CEL and adaptation to secondary CEF is possible, but the attenuation mechanisms, process for optimal attenuation, and feasible production of 63140 in tissue culture need to be further investigated, as well as the genomic basis for attenuation using whole genome sequencing technologies.

**Supplementary Materials:** The following supporting information can be downloaded at: https://www.mdpi.com/article/10.3390/poultry2040038/s1, Supplement Table S1. Summary of primers used for conventional PCR and real-time PCR.

**Author Contributions:** Conceptualization, G.Z. and M.G.; methodology, G.Z. and M.G.; software, G.Z., M.G., S.C., and V.A.P.-T.; validation, G.Z., and M.G. and S.C.; formal analysis, G.Z., M.G., S.C., and V.A.P.-T.; investigation, G.Z., M.G., and V.A.P.-T.; resources, G.Z. and M.G.; data curation, G.Z., M.G., and V.A.P.-T.; writing—original draft preparation, V.A.P.-T.; writing—review and editing, M.G.; visualization, G.Z., M.G., and V.A.P.-T.; supervision, G.Z. and M.G.; project administration, G.Z. and M.G.; funding acquisition, G.Z. and M.G. All authors have read and agreed to the published version of the manuscript.

**Funding:** This research received no external funding.

**Institutional Review Board Statement:** All animal experiments conducted in this study were performed under the Animal Use Proposal A2015 05001-Y3-A0 approved by the Animal Care and Use Committee (IACUC) in accordance with regulations of the Office of the Vice President for Research at the University of Georgia.

**Informed Consent Statement:** Not applicable.

**Data Availability Statement:** Data are contained within the article and Supplementary Materials.

**Acknowledgments:** The authors gratefully acknowledge Rodrigo Espinoza, Peter O'Kane, Denise Brinson, and Yun-Ting Wang for their collaboration during the execution of this project as well as Roy D. Berghaus for his valuable statistical insights. Special thanks to Mary Ard, EM Lab coordinator at the Georgia Electron Microscopy Facility at the University of Georgia for her help with the Electronic Microscopy protocols and images.

**Conflicts of Interest:** The authors declare no conflict of interest. During the course of this research, none of the authors held positions at companies other than the University of Georgia.

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
