# Peer review of "Attenuation of a Field Strain of Infectious Laryngotracheitis Virus in Primary Chicken Culture Cells and Adaptation to Secondary Chicken Embryo Fibroblasts"

_poultry, doi:10.3390/poultry2040038_

Round 1

Reviewer 1 Report

Comments and Suggestions for Authors

This manuscript reports the attenuation of a field strain of infectious laryngotracheitis virus(ILTV) in primary chicken culture cells and its adaptation to secondary chicken embryo fibroblast, which supplies a new understanding on the replication and pathogenicity of ILTV in cells and chickens. The study can arouse some interest and concern. But there are some suggestions in the design and results that need to be considered. 

1. In the results on the CPE of ILTV, is there any duplication? What is the number of duplicate samples? Please provide them in the methods or results.

2. In Table 1, The titers (TCID50) of ILTV should also be more repeatedly detected (N6) and the data should be analyzed using statistical analysis methods.

3. On the experiment of virus infected chickens, the further explanation on the scoring standard for pathological changes should be provided on in methods; and add the photographs of the chickens’clinical lesions as data supplements to the results.

4. In Figure 6 and In Figure 7, the statistical analysis and differential labeling of the data cannot reflect the changes in virulence of ILTV after passage. It is recommended to also perform statistical analysis on the lesion scores between all the ILTV infected groups, and labeling results in the figures.

5. There are some non-standard writing expressions in MS, such as CO2 (Page 3 Line128) should be CO2; º C (Page 3 Line 128 and 131) should be ℃, please check them and correct them.

Author Response

Thank you very much for your revision. 

  1. In the results on the CPE of ILTV, is there any duplication? What is the number of duplicate samples? Please provide them in the methods or results. - A brief explanation of the technique was added from 141-146. No duplicate titration was done. 

  1. In Table 1, The titers (TCID50) of ILTV should also be more repeatedly detected (N≥6) and the data should be analyzed using statistical analysis methods. Data was statistically analyzed following Modified Karber method for obtaining SE and later statistics. This was added to the paper on lines 145-146; 249-251; and 268-273. Modifications to Table 1 were done adding 95% CI, and statistical groups. 
  1. On the experiment of virus infected chickens, the further explanation on the scoring standard for pathological changes should be provided on in methods; and add the photographs of the chickens’clinical lesions as data supplements to the results.  Further explanation to the scoring standard for pathological changes was added from 227-238. Unfortunately, I lack of all photos for showing the whole array of chicken clinical lesions so I was not able to create a data supplement chart with photos for the results. 

  1. In Figure 6 and In Figure 7, the statistical analysis and differential labeling of the data cannot reflect the changes in virulence of ILTV after passage. It is recommended to also perform statistical analysis on the lesion scores between all the ILTV infected groups, and labeling results in the figures.

         Thank you for your observation. This was added as Fig 8, and commented from 454-456. 

  1. There are some non-standard writing expressions in MS, such as CO2 (Page 3 Line128) should be CO2; º C (Page 3 Line 128 and 131) should be℃, please check them and correct them. Thank you for this suggestion. This was addressed in all instances. 

Thank you very much for your suggestions. You have enriched this paper. Best, 

Reviewer 2 Report

Comments and Suggestions for Authors

The manuscript “Attenuation of a field strain of infectious laryngotracheitis virus in primary chicken culture cells and adaptation to secondary chicken embryo fibroblasts” is interesting and innovative, but there are some problems, which need to be carefully revised. This reviewer considers that the article can be accepted after addressing the following comments.

1. Figure 1, 2, 4: scale bar missing entirely.

2. Figure 4: Please provide the corresponding figure in bright field.

Author Response

Thank you very much for your comments. 

  1. Figure 1, 2, 4: scale bar missing entirely. Original photos were taken without a scale bar. I tried using Image J software to generate these scale bars, but I realized I could only approximate a scale bar and approximations can be mistaken. I have added a note on each photo indicating a 100X magnification.
  2. Figure 4: Please provide the corresponding figure in bright field. I was surprised I could find these photos on my notes - they are not the best. Please, kindly let me know if these would be satisfactory. 

Thank you for your review. This is a better paper thanks to your review.  

Best, 

Reviewer 3 Report

Comments and Suggestions for Authors

This study aimed to attenuate wild strains of ILTV by passaging by primary cells. The results showed that the virulence of ILTVs serially passage in CEL and CEF was significantly attenuated after P80 in CEL and after P10 in CEF. This attenuation is supported by animal studies, which are quite robust and provide important insights in terms of cellular attenuation of ILTV. However, some parts are difficult to understand, could be better organized and stated throughout, and are commented on below.

Comment 1

The study is very complete, with various approaches to ILTV attenuation and its evaluation throughout. However, as much of the supplementary information is contained in the text, it should be described in an organised manner throughout. For example, the experiments for quality control in Sections 2.5, 2.6 and 2.7 have little relevance to the main purpose of the study and are barely mentioned in the discussion, and should therefore be made into supplementary data or even briefly mentioned somewhere.

Comment 2

The viral strain description in section 2.1 lacks clarity. When mentioning that the 63140 strain underwent eight passages in CK cells, is it implied that this 63140-P8-CK was then used for subsequent experiments? The elaboration in Table 1 confuses the reader further. It might be more suitable to shift this portion to the "Serial Passages" or "Results" section. Incorporating a graphical representation of the experimental system could enhance comprehension.

Comment 3

Have you conducted whole-genome analyses for the passaged viruses? Comparing whole genomes might help elucidate genes associated with the virus's adaptation to host cells and virulence. If addressing this is challenging currently, perhaps it could be added as a recommendation for future studies.

Comment 4

The simple question is, why did you not compare more straightforwardly?

It is felt that it would make more sense to carry out a succession of ILTVs in CEK, CEL and CEF respectively and compare the results. In particular, the succession in the CEF is done after 52 generations of succession in the CEL, which may make it impossible to assess the original attenuation effect. Passage levels were also changed during the course of the experiment and, as the author also mentioned, it is felt that this change had a significant impact on the experiment. Furthermore, the changes appear to have had a negative impact on the animal experiment.

Comment 5

Would a comparison between CEL P60 and CEF P6 (CEL P52 + CEF P6) not be essential to gauge the effect of employing secondary CEF?

Minor Comments:

l   The line number on line 281 seems misplaced.

l   There are inconsistencies with the usage of superscript and subscript characters.

l   Are the Figure legends in Fig 3 accurate?

l   Lines 237 to 267 are not sub-sections, is there a reason for this?

Author Response

Thank you very much for your comments.

Comment 1

The study is very complete, with various approaches to ILTV attenuation and its evaluation throughout. However, as much of the supplementary information is contained in the text, it should be described in an organized manner throughout. For example, the experiments for quality control in Sections 2.5, 2.6 and 2.7 have little relevance to the main purpose of the study and are barely mentioned in the discussion, and should therefore be made into supplementary data or even briefly mentioned somewhere.

I agree they have little relevance to the main purpose. However, ILTV serial passage in CEF was quite rare in the literature, so we received a lot of skepticism from my department about the results which is why we went to this extend to make sure we addressed all opinions. 

Comment 2

The viral strain description in section 2.1 lacks clarity. When mentioning that the 63140 strain underwent eight passages in CK cells, is it implied that this 63140-P8-CK was then used for subsequent experiments? The elaboration in Table 1 confuses the reader further. It might be more suitable to shift this portion to the "Serial Passages" or "Results" section. Incorporating a graphical representation of the experimental system could enhance comprehension.

You are right. Further explanation has been added under lines 96-97. Table 1 was modified under CEF title to reflect this. Please, let me know if you still think a graphical representation is needed. 

Comment 3

Have you conducted whole-genome analyses for the passaged viruses? Comparing whole genomes might help elucidate genes associated with the virus's adaptation to host cells and virulence. If addressing this is challenging currently, perhaps it could be added as a recommendation for future studies.

This has been suggested and Dr. Garcia is looking into performing WGS in some passages to better understand attenuation. This has been added to lines 585-586

Comment 4

The simple question is, why did you not compare more straightforwardly?

It is felt that it would make more sense to carry out a succession of ILTVs in CEK, CEL and CEF respectively and compare the results. In particular, the succession in the CEF is done after 52 generations of succession in the CEL, which may make it impossible to assess the original attenuation effect. Passage levels were also changed during the course of the experiment and, as the author also mentioned, it is felt that this change had a significant impact on the experiment. Furthermore, the changes appear to have had a negative impact on the animal experiment.

Great comment. In short, this was a pet project during my Master's. Finding of CEF ILTV-CPE was completely out of the blue and worth pursuing. Passages were accelerated because my Master graduation was close and I wanted to test at least 100 passages in CEL, so that was the reason for accelerating incubation times. I still find valuable to communicate this in this format as attenuation for ILTV is not well understood. 

Comment 5

Would a comparison between CEL P60 and CEF P6 (CEL P52 + CEF P6) not be essential to gauge the effect of employing secondary CEF?

Great suggestion. An indirect comparison can be done using Fig 8 where all groups have been statistically analyzed together, and an indirect comparison can be made. This is discussed under lines 541-543. 

Minor Comments:

l   The line number on line 281 seems misplaced. It does not appear to be from my copy. Perhaps is edition, I'll keep my eyes open on the proof version. 

l   There are inconsistencies with the usage of superscript and subscript characters. This was addressed across the document, specially with the Log 10, CO2, and Celcius temperatures. 

l   Are the Figure legends in Fig 3 accurate? This was corrected. Please check Fig 3 modifications. 

l   Lines 237 to 267 are not sub-sections, is there a reason for this? No. I added the subtitle "Serial passaging in primary chicken cells". 

Thank you very much for your comments. Your review has enriched this paper. Thanks. 

Round 2

Reviewer 1 Report

Comments and Suggestions for Authors

Thanks for the authors' replies or responses to my concern questions, the manuscrip has been improved greatly and is complete although some leisions' photoes can not supplied.

Reviewer 3 Report

Comments and Suggestions for Authors

One point, the spp. in Mycoplasma spp. are not in italics, and there are some notational errors that could be corrected in the proof.

All other parts of the manuscript have been addressed in the comments and are more complete.

Excellent work!